# Etravirine Prevents West Nile Virus and Chikungunya Virus Infection Both In Vitro and In Vivo by Inhibiting Viral Replication

**DOI:** 10.3390/pharmaceutics16091111

**Published:** 2024-08-23

**Authors:** Xu Zheng, Yanhua He, Binghui Xia, Wanda Tang, Congcong Zhang, Dawei Wang, Hailin Tang, Ping Zhao, Haoran Peng, Yangang Liu

**Affiliations:** 1Department of Microbiology, Faculty of Naval Medicine, Naval Medical University, Shanghai 200433, China; zhengxu87@nankai.edu.cn (X.Z.); yanhua0556@163.com (Y.H.); xbhnjucpu@163.com (B.X.); wandt22@163.com (W.T.); luxion0227@gmail.com (C.Z.); grabbydowa@sina.com (D.W.); hailint@163.com (H.T.); pnzhao@163.com (P.Z.); 2Key Laboratory of Biological Defense, Ministry of Education, Naval Medical University, Shanghai 200433, China

**Keywords:** arboviruses, West Nile virus, chikungunya virus, FDA-approved reverse transcriptase inhibitor, antiviral drug, viral replication

## Abstract

Diseases transmitted by arthropod-borne viruses such as West Nile virus (WNV) and chikungunya virus (CHIKV) pose threat to global public health. Unfortunately, to date, there is no available approved drug for severe symptoms caused by both viruses. It has been reported that reverse transcriptase inhibitors can effectively inhibit RNA polymerase activity of RNA viruses. We screened the anti-WNV activity of the FDA-approved reverse transcriptase inhibitor library and found that 4 out of 27 compounds showed significant antiviral activity. Among the candidates, etravirine markedly inhibited WNV infection in both Huh 7 and SH-SY5Y cells. Further assays revealed that etravirine inhibited the infection of multiple arboviruses, including yellow fever virus (YFV), tick-borne encephalitis virus (TBEV), and CHIKV. A deeper study at the phase of action showed that the drug works primarily during the viral replication process. This was supported by the strong interaction potential between etravirine and the RNA-dependent RNA polymerase (RdRp) of WNV and alphaviruses, as evaluated using molecular docking. In vivo, etravirine significantly rescued mice from WNV infection-induced weight loss, severe neurological symptoms, and death, as well as reduced the viral load and inflammatory cytokines in target tissues. Etravirine showed antiviral effects in both arthrophlogosis and lethal mouse models of CHIKV infection. This study revealed that etravirine is an effective anti-WNV and CHIKV arbovirus agent both in vitro and in vivo due to the inhibition of viral replication, providing promising candidates for clinical application.

## 1. Introduction

Arboviruses are the etiologic agents of many incapacitating diseases that can progress to severe and lethal forms, affecting the human population worldwide; therefore, they are considered a global health problem by the World Health Organization (WHO) [1]. Among them, the West Nile virus (WNV) and chikungunya virus (CHIKV) are widespread arboviruses, resulting in a large number of clinical patients per year [2,3]. WNV was discovered on the African continent in 1937 and reemerged in the summers of 2012, 2016, and 2018 in the USA and Europe, causing an unusual epidemic of WNV, during which more than 280 people died [4]. Since the first reports of CHIKV infection in Africa in the 1950s, subsequent epidemics of CHIKV occurred throughout the latter half of the 20th century in countries within Asia and sub-Saharan Africa [5]. Since 2006, it has spread to new areas, causing disease on a global scale, and the potential for CHIKV epidemics remains high [6,7]. The epidemics are caused by a single amino acid substitution in the E1 glycoprotein (A226V); this variant (CHIKV LR2006 OPY1) has been shown to increase replication and transmission in *Aedes. albopictus* mosquitoes and is important for cholesterol-dependent entry [8,9]. Unfortunately, there is an unmet need to develop specific antiviral therapeutics for WNV and CHIKV.

WNV is a member of the Flavivirus genus—other members of this genus include Zika virus (ZIKV), tick-borne encephalitis virus (TBEV), yellow fever virus (YFV), Japanese encephalitis virus (JEV), and dengue virus (DENV)—and is considered one of the most important causative agents of human viral encephalitis [4,10]. WNV transmission occurs between mosquitoes (e.g., *Culex tarsalis* and *Culex pipiens*) and various bird species (e.g., *Turdus migratorius*) in nature [11]. After the virus successfully invades humans through WNV-infected mosquito bites, it replicates in multiple target tissues and cells, such as vascular endothelial cells, neurons, and hepatocytes, and causes fever, headache, viremia, and neurological disorders in 20% of infected patients; these symptoms persist for days to weeks [12,13,14].

CHIKV belongs to the genus Alphavirus of the family Togaviridae and spreads through the bite of infected mosquitoes (e.g., *Aedes stegomyia*) and is responsible for periodic outbreaks of febrile disease. Unlike WNV, a characteristic feature of CHIKV disease is recurring musculoskeletal disease primarily affecting the peripheral joints that can persist for months to years after acute infection [15,16,17]. CHIKV disease is often self-limiting and has a low fatality rate but has a high fatality and disability rate due to CHIKV-associated central nervous system (CNS) disease in infants and elderly individuals [18,19].

RNA-dependent RNA polymerase (RdRp) is one of the largest and most conserved proteins among the RNA viral, and is vital for viral replication [20]. In flavivirus, nonstructural protein 5, containing an RdRp domain, has evolved with diverse mechanisms to suppress host immune signaling to promote viral infection, targeting signal transducers of transcription 2 (STAT2) and heat shock protein 90 (HSP90) [21,22,23]. It is considered one of the most intriguing and promising targets for antiviral drug development [24,25,26]. RdRp inhibitors have been launched or are currently in clinical trials for the treatment of flavivirus infections [27,28]. Some compounds targeting the RdRp of WNV, such as favipiravir T-705 and sofosbuvir, also illustrated inhibition of the replication of CHIKV and other alphaviruses [29,30].

Drug repurposing is an important strategy for discovering clinical candidates for emerging viruses [31,32,33,34]. Since there are a variety of reverse transcriptase inhibitors used in the clinical treatment of HIV infection, some reverse transcriptase inhibitors can also effectively inhibit RNA viruses such as azvudine and ritonavir [35,36]. In this study, 27 FDA-approved reverse transcriptase inhibitors were repurposed for the assessment of anti-arboviruses. Four compounds—etravirine, salicylanilide, efavirenz, and rilpivirine—were found to significantly suppress WNV infection. Further investigation revealed that etravirine also inhibited YFV, TBEV, and CHIKV. Moreover, etravirine also displayed potent anti-WNV and anti-CHIKV activity in vivo.

## 2. Materials and Methods

### 2.1. Cell Lines, Viruses, and Compounds

SH-SY5Y cells, HUVECs, and Vero cells were purchased from ATCC (Mansas, VA, USA), and the Huh7 cell line was purchased from the Chinese Academy of Sciences (Shanghai, China). The cells were maintained in Dulbecco’s modified Eagle’s medium (DMEM, Thermo Fisher Scientific, Waltham, MA, USA) (SH-SY5Y, Vero, and Huh 7) or endothelial cell medium (ScienCell, San Diego, CA, USA) (HUVEC) supplemented with 10% fetal bovine serum (FBS, Gibco) and with endothelial cell growth factor (ECGF) in a humidified incubator at 37 °C (5% CO_2_). An FDA-approved reverse transcriptase inhibitor library comprising 27 compounds was purchased from Selleck Chemicals (Selleck, Houston, TX, USA). All compounds were dissolved in dimethyl sulfoxide (DMSO) as a 10 mM stock solution and stored at −80 °C. The WNV NY2000 strain (AF404756), TBEV Zmeinogorsk-5 strain (KY069125), and CHIKV LR2006 OPY1 (DQ443544) strain were synthesized in this laboratory. The YFV YFV0 strain was isolated from a confirmed YF patient (FJYF03/2016, GenBank: KY587416.1). All animal experiments were approved by the Institutional Committee for Biosafety and Animal Care of Naval Medical University and were performed in an animal BSL-3 laboratory.

### 2.2. Drug Inhibition Assay and IC50 Calculation

The stock solution (10 mM) was diluted in DMEM containing 2% FBS to generate a working solution at different concentrations. For screening, the FDA-approved reverse transcriptase inhibitor library comprising 27 compounds (Appendix A) in DMSO was diluted to form a 25 μM working solution. For dose-response studies, etravirine was serially diluted 2-fold to final concentrations ranging from 50 μM to 1.625 μM. The same volume of DMSO was used as a negative control.

Huh7 cells, SH-SY5Y cells, and HUVECs were seeded separately in 96-well plates and cultured to 90% confluence. WNV, TBEV, CHIKV, or YFV were added to the plate at an MOI of 0.1 and incubated at 37 °C for 2 h. After being washed with PBS, the infected cells were treated with a drug-working solution for 24 h at 37 °C before being fixed with cold methyl alcohol or lysed with TRIzol. Then, an immunofluorescence (IF) assay (see below) was performed to detect the infected cells, and real-time quantitative PCR (RT–qPCR) was performed as described below to determine the viral RNA load. The half-maximal inhibitory concentration (IC50) of etravirine on virus infection was calculated using the nonlinear regression model in GraphPad Prism 7.00 (GraphPad Software, San Diego, CA, USA). In parallel, uninfected cells were treated with drugs via the same method, and drug toxicity was assessed as described below.

### 2.3. Time-of-Drug-Addition Assay

To assess the effects of etravirine on the WNV life cycle, a time-of-drug-addition assay was conducted as described previously [37]. Huh-7 cells were grown on 96-well plates and allowed to reach 90% confluence. Etravirine (25 μM) or DMSO was added 2 h before (pre-2-0), simultaneously with 0 (0–2), or 2 (2–4), 4 (4–6), 6 (6–8), 8 (8–10), 10 (10–12), 12 (12–14), or 14 (14–24) hours after virus infection, and the mixture was incubated for 2 h (the action period of 14–24 h was 11 h) before washing and adding fresh medium. Twenty-four hours post-infection (24 hpi), the plates were fixed with cold methanol and IF assays were performed as described below.

### 2.4. Immunofluorescence Assay

IF assays were conducted as described previously [38]. Briefly, after treatment, the cells were fixed with cold methanol at −20 °C for 25 min and blocked with 3% bovine serum albumin (BSA, 2 h) at room temperature. Then, the cells were incubated with primary antibodies (showed in Appendix A) against the virus (overnight, 4 °C). After being washed twice, the cells were stained at room temperature for 2 h with AF-488-conjugated donkey anti-rabbit IgG (Thermo Fisher Scientific, San Diego, CA, USA) and for an additional 10 min with DAPI (Sigma–Aldrich, St. Louis, MO, USA) for nuclei staining. Images were captured with a Cytation 5 (Biotek Instruments, Santa Clara, CA, USA). The infected (with fluorescence intensity more than 5000) and total cells were counted using Gen5 3.10 software in Cytation 5, and the percentage of infected cells (infectivity) was normalized to that of the control group and illustrated in GraphPad Prism 7.00.

### 2.5. Molecular Docking

The crystal structure of WNV RdRp (PDB: 2HCN) and alphavirus RdRp (nsP4) (PDB: 7VW5) were downloaded from PDB database (https://www.rcsb.org/; accessed on 10 April 2024) and prepared by removing water and adding hydrogen atoms with PyMOL 2.4.1 (https://pymol.org/2/; accessed on 10 April 2024), and repairing the structure using the protein preparation tool [39]. The 3D conformer of etravirine was downloaded from the Pubchem database (https://pubchem.ncbi.nlm.nih.gov/; accessed on 10 April 2024). Then, the Autodock 4 software was used for simulations for predicting binding affinities and poses between the drug and the RdRps [40]. The residues within a 10 Å radius around the ligand were defined as available. After docking was complete, the conformation with the lowest binding energy was selected for subsequent analysis. The best poses of the selected conformation were imaged and the distance between hydrogen bonds was measured using PyMOL 3.0 Version 3.0.4.

### 2.6. Cell Viability Assay

A Cell Counting Kit-8 (CCK-8, Beyotime, Suzhou, China) was utilized to assess the cell toxicity of the drugs following the manufacturer’s instructions. Briefly, Huh 7 cells, SH-SY5Y cells, and HUVECs were separately grown on 96-well plates and allowed to reach 90% confluence before treatment with different concentrations (described above) of etravirine or DMSO. Twenty-two hours after treatment, the cells were washed twice with PBS, stained with CCK-8 working solution (1:9 diluted in culture medium) for three hours at 37 °C, and scanned using a microplate reader (BioTek, Santa Clara, CA, USA) for absorption at 450 nm (OD450). The cell viability was calculated by the following Formula (1):Cell viability (% of control) = [OD450_drug_ − OD450_PBS_]/[OD450_DMSO_ − OD450_PBS_] × 100(1)

### 2.7. Real-Time Quantitative PCR (RT–qPCR)

Treated Huh7 cells, SH-SY5Y cells, HUVECs, or mouse tissues were lysed using TRIzol (TaKaRa, Shiga, Japan), and total RNA was extracted following the manufacturer’s instructions. A PrimeScript RT Master Mix kit (TaKaRa) was used to synthesize cDNA, and SYBR Premix Ex Taq (TaKaRa) was used to perform PCR amplification of WNV RNA, inflammatory cytokines, and the internal reference gene glyceraldehyde-3-phosphate dehydrogenase (GAPDH), using the appropriate primer pairs (Appendix A) in an ABI 7300 system (Applied Biosystems, Carlsbad, CA, USA). The relative quantity (RQ) of target genes, such as WNV RNA and inflammatory cytokine mRNAs, was calculated by normalization against the GAPDH levels and no-drug-treatment infection group using comparative cycle threshold values (2^−ΔΔCt^) as follows (2):RQ_target_ = 2^−(drug[CTtarget − CTGAPDH] − blank[CTtarget − CTGAPDH])^(2)

### 2.8. Replication Determination

For WNV, a genome-transient expression assay was performed to assess its replication. The viral RNA of WNV was isolated from virions using a QIAamp Viral RNA Mini Kit (Qiagen, Hilden, NRW, Germany) and transfected into Huh7 cells grown on 24-well plates using Lipofectamin^TM^ 2000 (Lipo2000). Six hours post-transfection (6 hpt), the supernatant was replaced with fresh medium containing different concentrations of etravirine for another four hours of incubation (10 hpt). Before the cells were lysed by TRIzol, the WNV RNA load in both the supernatant and cells was analyzed by RT–qPCR. The RQ of WNV RNA compared with that of the DMSO-treated group represents the replication level of WNV at 10 hpt.

For CHIKV and YFV, replicon plasmids of the CHIKV LR2006-OPY1 strain or YFV-17D strain carrying the enhanced green fluorescent protein (EGFP) or nanoluciferase (NanoLuc) reporter gene were constructed as described (Appendix A). After linearization with *NotI* (for the CHIKV-LR2006 replicon) or *SmaI* (for the YFV-17D replicon), these plasmids were used as templates for viral replicon RNA transcription using the HiScribe T7 ARCA mRNA Kit (New England Biolabs, Ipswich, MA, USA). Then, the RNA transcripts were transfected into Vero or Huh7 cells with Lipo2000 transfection reagent (Invitrogen, San Diego, CA, USA) according to the manufacturer’s instructions for six hours before the supernatant was replaced with fresh medium containing 25 μM etravirine or DMSO. Twenty-four hours post-transfection (24 hpt), the expression of reporter genes was detected as follows: the fluorescence of the CHIKV-EGFP replicon was imaged using Cytation 5, and the mean fluorescence intensity was tested by Gen5 3.10 and normalized to that of the DMSO-treated control; after incubation with 10% NanoLuc reagent for 5 min in the dark, the luminescence of NanoLuc (CHIKV-LR2006-NanoLuc and YFV-17D-NanoLuc) was captured using a chemiluminescence imaging system (Clinx Science Instruments, Shanghai, China).

### 2.9. In Vivo Experiment

To detect anti-WNV activity, six-week-old C57BL/6 mice were divided randomly into 3 groups (16 mice per group) and inoculated via intraperitoneal (i.p.) injection with 10^4^ plaque-forming units (PFUs) of WNV or PBS followed by oral administration of DMSO (solvent control) or etravirine in corn oil, with etravirine dose of 40 mg/kg/days as described [41]. The drugs (including DMSO in corn oil) were continued daily for 5 days (from the days before infection to 3 days post-infection, −1–3 dpi). The survival and body weight of the mice were monitored daily for 16 days post-WNV inoculation. On the 4th and 6th days post-infection (4 dpi and 6 dpi), brains were obtained from 3 sacrificed mice per group, lysed with TRIzol for virus load and inflammatory cytokine evaluation via RT–qPCR (see above), or ground in cold PBS for assessment of viral titer via plaque assay [42,43].

To assess the anti-CHIKV activity, six-week-old C57BL/6 mice were divided randomly into 3 groups (13 mice per group). For the lethal model, mice were infected via intranasal injection with CHIKV ROSS strain (10^5^ PFU) or PBS. For the nonlethal model, mice were injected subcutaneously with CHIKV LR2006 OPY1 (10^3^ PFU) or PBS into the left rear feet. The infected mice were then treated by oral administration of DMSO in corn oil (solvent control) or etravirine (40 mg/kg) in corn oil. The drugs were continued as described in the WNV treatment (above). The survival and body weight of the lethal model groups were monitored daily for 16 days. At the same time, the relative change in the footpad swelling of the nonlethal model groups was measured daily for 10 days. At 4 dpi, brains (from lethal model groups) or muscle near the footpad (from nonlethal model groups) were obtained from 3 sacrificed mice per group, lysed, and virus load/titer, and inflammatory cytokine expression were analyzed as described in WNV treatment.

As needed based on illness, after overall assessment of moribund animal, inability to move to food or water, weight loss and so on, infected mice were euthanized via isoflurane overdose to reduce suffering, which complied with the relevant ethical regulations.

### 2.10. Statistical Analysis

The measurement data are displayed as the arithmetic mean ± standard deviation (SD), and the significant differences between groups were evaluated using GraphPad Prism 7.00 (GraphPad Software, San Diego, CA, USA). Student’s *t*-test was used for two unpaired groups, and one-way ANOVA followed by the Bonferroni post hoc correction was used for multiple comparisons between multiple groups. Each experiment was repeated at least three times.

## 3. Results

### 3.1. Screening of FDA-Approved Reverse Transcriptase Inhibitors for WNV Infection

To investigate the potential of FDA-approved reverse transcriptase inhibitors for being repurposed as anti-WNV agents, a library comprising 27 FDA-approved reverse transcriptase inhibitors was utilized to screen for WNV infection in vitro. Huh7 and SH-SY5Y (SY5Y) cells were incubated with WNV (MOI = 0.1) for 2 h, followed by treatment with the small molecule library (25 μM). Twenty-four hours post-infection (24 hpi), the drugs were removed, and an immunofluorescence (IF) assay was performed to detect WNV infection. Four out of the twenty-seven compounds—etravirine, salicylanilide, efavirenz, and rilpivirine—exhibited significant antiviral activity (inhibitory rate ≥ 50%, *p* < 0.001) against WNV in both cell lines, especially etravirine and rilpivirine, which inhibited approximately 84.92% and 85.10%, respectively, in Huh 7 cells (Figure 1A) and 82.57% and 75.71%, in SY5Y cells (Figure 1B). Moreover, there was no significant difference in the inhibitory effect of etravirine or rilpivirine between the two types of cells. The above results show that etravirine and rilpivirine exhibit comparable anti-WNV efficacy and that their antiviral effects are independent of cell type, although mildly lower antiviral efficiency against SY5Y cells was observed than against Huh7 cells, which needs further confirmation. In addition, though salicylanilide and efavirenz show more than 50% inhibitory rate for WNV infection (*p* < 0.001), their efficacy is significantly lower than that of etravirine and rilpivirine. Some other NNRTIs, such as tenofovir (including its derivant), didanosine, and zidovudine, inhibited the infection with an efficiency of less than 50% (*p* < 0.05). Rilpivirine has been reported to inhibit the enzymatic activity of NS5 and suppress ZIKV infection and replication, but anti-flavivirus effects of etravirine have not been reported. Therefore, in this study, the antiviral efficacy and mechanism of etravirine were further evaluated both in vitro and in vivo.

### 3.2. Etravirine Has Dose-Dependent and Cell-Type-Independent Effects on WNV

To assess the activity of etravirine on WNV invading its target tissues, we determined the dose-response of WNV infection to etravirine in human cell lines Huh 7 and SY5Y and in primary human umbilical vein endothelial cells (HUVECs) derived from the above target tissues. After exposure to WNV, the cells were incubated with different doses (six 2-fold serial dilutions, 50 μM to 1.625 μM) of etravirine. Viral protein and RNA loads were detected by IF and RT–qPCR, respectively, at 24 hpi, and the half-maximal inhibitory concentration (IC50) of etravirine was calculated. Moreover, the cell toxicity at different doses of drugs was characterized using a Cell Counting Kit-8 (CCK-8) assay. As shown in Figure 2A–C, the infectivity of WNV in the three cell types decreased with increasing the etravirine dose. The IC50s of etravirine on WNV infection in Huh7 cells, SY5Y cells, and HUVECs were 3.3 μM, 6.3 μM, and 3.6 μM, respectively.

Moreover, the viral RNA load was also inhibited by etravirine in a dose-dependent manner in all target cells (Figure 2D–F), which is consistent with the infectivity results, indicating that etravirine has a dose-dependent effect on WNV invasion of cells derived from various target tissues. Cell viability assays revealed that 0–25 μM etravirine had no obvious toxicity to all the cells, while at the 50 μM concentration, it was slightly toxic to SY5Y cells with a decrease in cell viability of approximately 21.8% (*p* < 0.001) (Figure 2A–C). Since infection of the virus with this concentration of etravirine may interfere with cell damage signals, we selected 25 μM as the working concentration of etravirine for subsequent research. Interestingly, from the IC50 results, we found that the antiviral effect of etravirine on SY5Y cells was also weaker than that on the other two cells, in line with the phenomenon shown in Figure 1, indicating that there may be factors interfering with the effect of etravirine on SY5Y cells.

### 3.3. Etravirine Exhibits Broad Anti-Arboviral Activity

To determine the therapeutic effect of etravirine on other arboviruses, we evaluated the dose-response effects of etravirine on other flaviviruses, such as YFV and TBEV, and alphaviruses, such as CHIKV. After infection with the above viruses, Huh7 cells, SY5Y cells, and HUVECs were treated with different concentrations of etravirine, and viral infectivity was assessed using IF assays. As for WNV, etravirine had a dose-dependent inhibitory effect on YFV infection in various target cells, with IC50 values of 3.9 μM, 7.0 μM, and 3.1 μM in Huh7 cells, SY5Y cells, and HUVECs, respectively (Figure 3A,D,G). As expected, its ability to inhibit YFV infection in SY5Y cells was weaker than that in the other two cell lines. This phenomenon differed among CHIKV-infected Huh7 cells, SY5Y cells, and HUVECs, with nearly the same IC50 values of 9.9 μM, 9.6 μM, and 9.6 μM of etravirine, respectively (Figure 3C,F,I). These results also suggest that although etravirine has dose-dependent inhibitory effects on CHIKV infection, its efficacy is not as high as that on YFV and WNV. Interestingly, in terms of TBEV infection, etravirine dose-dependently suppressed TBEV infection in Huh7 cells (IC50 = 6.9 μM) and HUVECs (IC50 = 8.0 μM) (Figure 3B,H), but it had no obvious effect on TBEV infection in SY5Y cells (*p* > 0.05) (Figure 3E). These results reveal that even though etravirine has a broad protective effect against multiple arboviruses in target cells, its potential for improving the neuroinvasion and pathogenicity of TBEV may be limited. Moreover, the inhibitory effect of etravirine on TBEV infection of peripheral target cells (hepatocytes and vascular endothelial cells) was not as strong as that on YFV and WNV infection.

### 3.4. Etravirine Inhibits Viral Replication

To further explore the effect of etravirine on the viral life cycle, a time-of-drug-addition assay was subsequently performed. Briefly, 2 h of treatment with etravirine (25 μM) was added to different periods of WNV (MOI = 0.1) infection: 2 h before WNV infection (pre-2-0 h), coincubation with the virus (0–2 h), and 2 h (2–4 h), 4 h (4–6 h), 6 h (6–8 h), 8 h (8–10 h), 10 h (10–12 h), 12 h (12–14 h), and 14 h (14–24 h, long-term treatment) after viral incubation. An IF assay was utilized to determine the infectivity at 24 hpi. As shown in Figure 4A, etravirine exhibited prominent suppressive activity during 4 h to 12 h of WNV infection, with an inhibition efficiency greater than 75%. Although treatment with etravirine at the early (0–4 h, which may be the entry stage) or the latest stage (12–14 h, which may be the release stage) affects WNV infection, with an inhibitory rate of less than 50%, these results suggest that etravirine mainly acts in the middle stages of the viral life cycle, possibly through the replication process.

Since there is no WNV replicon available, to confirm the effect of etravirine on its replication, a genome transient expression assay was carried out. To avoid the impact of the entry process, the whole genome of WNV was isolated from virions and directly transfected into Huh 7 cells, and the viral RNA load in the cells and culture media was assessed by RT–qPCR after treatment with different concentrations of etravirine for 10 h post-transfection (10 hpt). As shown in Figure 4B, the viral RNA load in cells was inhibited by etravirine in a dose-dependent manner, and 45 μM etravirine nearly completely blocked WNV RNA amplification. However, viral RNA in the culture medium was undetectable at 10 hpt in any group, revealing little release of the virus at this time point. These results confirmed that etravirine suppressed the WNV replication stage. To confirm this effect on other viruses, we constructed single-round CHIKV-LR2006 and YFV-17D replicon plasmids carrying enhanced green fluorescent protein (EGFP) or nanoluciferase (NanoLuc) reporter genes. The CHIKV-LR2006 replicon RNA transcribed from the linearized plasmid was transfected into Vero or Huh 7 cells for 6 h before treatment with etravirine (25 μM), and the expression of the replicon reporter gene EGFP was imaged at 24 hpi. As shown in Figure 4C, compared with Vero cells, etravirine significantly inhibited the expression of EGFP carried by the CHIKV-LR2006 replicon, especially in Huh7 cells (inhibition rate of approximately 89%, *p* < 0.001) (inhibition rate of approximately 59%, *p* < 0.001). At the same time, single-round CHIKV-LR2006 and YFV-17D replicon RNAs carrying NanoLuc were transfected into Huh7 cells, which were subsequently incubated with different doses of etravirine. As expected, by monitoring the activity of NanoLuc using luciferin 24 hpt, we found that etravirine inhibited CHIKV-LR2006 replicon expression in a dose-dependent manner, and 15 μM and 45 μM etravirine almost completely blocked its expression (Figure 4D) (*p* < 0.001). Surprisingly, in contrast to the infectivity results shown in Figure 3, although etravirine also dose-dependently suppressed the expression of the YFV-17D replicon, its efficiency was not as strong as that of the CHIKV-LR2006 replicon (Figure 4E). The above results indicate that etravirine acts as a replication inhibitor for multiple arboviruses.

### 3.5. Etravirine Has the Potential to Interact with RdRp

Due to the inhibitory effect of etravirine on CHIKV, YFV, and WNV replication, we speculated that this inhibitor might also function by targeting viral RdRp. Thus, molecular docking was subsequently performed to analyze the interplay between etravirine and the RdRp domain in WNV nonstructural protein 5 (NS5) (PDB: 2HCN) and alphavirus RdRp (nsP4) (PDB: 7VW5). As shown in Figure 5A, the 3D structural models of WNV RdRp and alphavirus RdRp in complex with etravirine indicated optimal structural quality. In the left panel of Figure 5A, etravirine interacted best with the pocket beside the active center of WNV RdRp, which plays versatile roles in viral RNA replication (ΔG: −36.401). Four potential hydrogen bonds (with distances of 1.8, 1.8, 2.1, and 2.2 Å) were formed between the macrolide skeleton of the etravirine and the WNV RdRp residues (Figure 5B, upper right panel), which means a strong and possibly covalent interaction. Since the crystal structure of CHIKV RdRp (nsP4) has not yet been reported and the RdRp of alphaviruses is highly conserved among the family members, alphavirus RdRp was used here to characterize its potential interaction with etravirine. As expected, etravirine could also interact with a pocket on the surface of the alphavirus RdRp (Figure 5A, right panel; Figure 5B, lower panel) (ΔG: −35.313). Similarly, four potential strong hydrogen bonds (with distances of 1.9, 2.0, 2.1, and 2.4 Å) were formed between etravirine and the alphavirus RdRp residue (Figure 5B, lower right panel). These results further elucidated the inhibitory effect of etravirine on flavivirus and alphavirus replication.

### 3.6. Etravirine Protects Mice from WNV Infection-Induced Lethality

To assess the in vivo antiviral potential of etravirine, we next evaluated its protective effects in mice challenged with WNV. First, to determine whether etravirine could rescue mice from WNV-induced death, six-week-old C57BL/6 mice inoculated intraperitoneally (i.p.) with a lethal dose of WNV were treated with either DMSO in corn oil (solvent control) or etravirine (40 mg/kg) in corn oil daily from the day before infection to the 3rd day post-infection (−1 to 3 dpi). Body weight and survival were monitored daily for 16 days (Figure 6A). From 4 to 5 dpi, all mice in the DMSO-treated group started to show neurological symptoms, including a hunched posture, ruffled fur, tremors, and impaired mobility. From 7 to 9 dpi, all the mice in this group succumbed to the infection (Figure 6B, the green line). However, treatment with etravirine (WNV + etravirine group) significantly relieved mice symptoms and shows a 50% protection rate against death in mice (*p* < 0.001) (Figure 6B, the orange line). The median survival time improved from 8 (DMSO) to 13 (etravirine) days. Consistently, etravirine treatment also protected mice from WNV infection-induced body weight loss (Figure 6C). At 4 and 6 dpi, the brains of the infected mice were isolated, and the viral load was assessed via plaque assays or RT–qPCR. As expected, repeated administration of etravirine dramatically decreased the number of live virions and the amount of viral RNA in the brain at both 4 and 6 dpi compared with those in the DMSO group (*p* < 0.05) (Figure 6D,E). This finding was further supported by the expression levels of inflammatory cytokines as mice treated with etravirine showed lower expression of *TNF-α*, *CCL-2*, and *IL-10* at 4 dpi and lower levels of *TNF-α*, *CCL-2*, and *IL-6* at 6 dpi than mice treated with DMSO (Figure 6F–I). In conclusion, etravirine inhibited WNV infection in vivo and attenuated inflammation induced by WNV.

### 3.7. Etravirine Protects Mice from CHIKV Infection-Induced Lethality and Footpad Swelling

To evaluate the in vivo anti-CHIKV effect of etravirine, we utilized two mouse models, as described in the Methods Section 2.9. For assessment of its protection of arthrophlogosis, mice were infected with the CHIKV LR2006 OPY1 strain via subcutaneous injection in the left rearfoot followed by etravirine (40 mg/kg) treatment for 5 days (Figure 7A). Mice were monitored for footpad swelling, viral load, and inflammatory cytokine levels. Injection of LR2006 OPY1 resulted in acute footpad swelling, especially in the DMSO-treated group, which peaked at 5 dpi. Compared with DMSO, etravirine alleviated the symptoms of swelling (Figure 7B). Additionally, virus loads in the left rear muscles of etravirine-treated mice were significantly lower than those in the left rear muscles of DMSO-treated mice at 4 dpi (Figure 7C,D). Similar to ROSS strain infection, the expression of *TNF-α*, *IL-6*, *CCL-2*, and *IL-10* was reduced upon etravirine treatment compared with DMSO treatment under the LR2006 OPY1 strain challenge. These data suggest the potential of etravirine to manage CHIKV infection and related symptoms and inflammation in vivo.

For rescue–lethal assessment, mice were challenged with the CHIKV ROSS strain via the intranasal route at a lethal dose and orally administered etravirine (40 mg/kg) or DMSO daily in corn oil for 5 days (Figure 8A). Upon being challenged with CHIKV-ROSS, most of the mice (60%) exhibited hunched posture, ruffled coat, and impaired mobility at 3 dpi. All the mice had impaired mobility at 5 dpi. In contrast, 60% of the etravirine-treated mice maintained a healthy state at the same time (Figure 8B and Appendix A). Infection resulted in severe disease symptoms in the DMSO-treated group, and all the CHIKV-infected mice died at 11 dpi. However, treatment with etravirine significantly attenuated disease symptoms, resulting in 30% survival (Figure 8C). Considering body weight, the DMSO-treated group exhibited and maintained extreme weight loss from 5 dpi until death (Figure 8D). Compared with DMSO treatment, etravirine treatment significantly reduced the weight loss rate, which gradually increased at 11 dpi. After that, the viral titer and amount of viral RNA in the brain were also estimated at 4 dpi. As expected, in the etravirine (CHIKV + etravirine) group, remarkable reductions in the viral titer and RNA were observed (Figure 8E,F). Infection with CHIKV also increased the levels of inflammatory cytokines and chemokines (*TNF-α*, *IL-6*, *IL-10*, and *CCL-2*), which are associated with severity, morbidity, and mortality in patients. Interestingly, the expression of *TNF-α*, *IL-6*, *CCL-2*, and *IL-10* was downregulated upon etravirine treatment compared with DMSO treatment (Figure 8G).

## 4. Discussion

The difficulty in preventing arbovirus global inundation represents an important public health problem. Among these viruses, West Nile virus (WNV) and chikungunya virus (CHIKV) are the most widespread mosquito-borne viruses, affecting a large number of clinical patients [2,3,4,10]. Recently, in the USA and Europe, WNV reemerged in the summers of 2012, 2016, and 2018, causing an unusual epidemic of WNV, during which more than 280 people died and over 50% developed neuroinvasive diseases [4,44]. In 2015, in the Americas, 37,480 confirmed CHIKV cases were reported to the Pan American Health Organization (PAHO) regional office; subsequently, in 2016, a total of 146,914 laboratory-confirmed cases were registered [45]. Unfortunately, no effective human vaccines or specific antiviral therapeutics are available to date. In comparison with novel drug development, approved drug repurposing is safer, less expensive, convenient, and has a rapid development period. In particular, the FDA-approved small-molecule regimen remains the most desirable treatment option or research foundation [46].

In this study, through screening with an FDA-approved reverse transcriptase inhibitor library and further verification, we unfolded that etravirine shows significant inhibition of the infection by multiple arboviruses, including WNV, YFV, TBEV, and CHIKV, in a dose-dependent manner. Further exploration of the phase of action shows that etravirine mainly acts at the stage of viral replication, possibly through binding to the RdRp of arboviruses. In vivo, etravirine protected mice from WNV and CHIKV-induced bodyweight loss, tissue injury, and death and reduced viral load and inflammatory factor expression in tissues. To the best of our knowledge, this is the first report on the effect of etravirine against arboviruses, especially on their replication. Our results show that etravirine may be a promising broad-spectrum anti-arbovirus agent for use against infections.

The primary objective of this study was to evaluate FDA-approved reverse transcriptase inhibitors as potential candidates with anti-WNV activity, both in vitro and in vivo. Four compounds, namely, etravirine, salicylanilide, efavirenz, and rilpivirine, inhibited WNV infection by more than 50% in Huh7 and SY5Y cells (Figure 1A,B). Salicylanilide has been reported to significantly inhibit various virus species, including coronaviruses, HCV viruses, herpes viruses, and HIV viruses [47]. Nitazoxanide, a salicylamide derivative, reportedly inhibits the invasion of WNV both in vitro and in vivo [48]. Studies have reported that rilpivirine inhibits the enzymatic activity of NS5 and suppresses ZIKV infection and replication in primary human astrocytes [49]. As a sensitive anti-HIV drug, efavirenz was also found to inhibit WNV infection but with significantly lower efficiency than rilpivirine (Figure 1A,B). Fortunately, etravirine, a non-nucleoside reverse transcriptase inhibitor (NNRTI), significantly inhibited WNV infection in a dose-dependent manner at micromolar concentrations with low cytotoxicity (Figure 2A,D). Cytotoxicity analysis indicated that etravirine has a large therapeutic window for WNV and even arboviruses. Besides, we also noticed in Figure 1 that some NNRTIs, such as tenofovir (including its derivant), didanosine, and zidovudine, also show antiviral ability with an efficiency of less than 50%. On the one hand, those small molecular agents may really exhibit anti-WNV activity, while their IC50 may be much higher than etravirine and ripivirine. On the other hand, the targets of those compounds may not be the key factor of WNV infection or there are complementary mechanisms to the targets. Thus, the potential for the application of these drugs in WNV is relatively small. According to the characteristics of HIV reverse transcription and the long-term treatment of patients, drug resistance in the application of NNRTI, including etravirine [50] is noteworthy. However, considering WNV characteristics and the therapeutic schedule of WNV-infected patients, etravirine use as a potential anti-WNV drug does not present an acute conflict.

After WNV enters the host through a mosquito bite, the virus replicates in keratinocytes, skin cells, and vein endothelial cells, and then, it arrives at and infects peripheral organs such as the liver and kidneys through blood and lymphoid circulation. As shown in Figure 2, etravirine significantly inhibited WNV infection in multiple cell lines, such as Huh7 cells, SY5Y cells, and HUVECs, revealing that etravirine might protect against WNV infection in patients who experience disease progression or develop multiorgan failure. It has been reported that over 50% of WNV-infected patients have neuroinvasive diseases that significantly increase the risk of death during acute hospitalization [4,44]. Etravirine potentially inhibits WNV infection in SY5Y cells, demonstrating that it might be effective at suppressing WNV invasion in the central nervous system, thus preventing potential neuroinvasive damage and decreasing patient mortality.

Moreover, the data presented in Figure 3, together with the previously published antiviral activity of etravirine against HIV and high binding energy with both Mpro and RdRp of SARS-CoV-2, indicate that etravirine possesses potential broad-spectrum antiviral activity [51,52]. We found that etravirine mainly functions at the post-entry step of the WNV viral life cycle, especially at 4 hpi to 12 hpi, indicating that it has a large therapeutic window to target various proteins of WNV or the host. This phenomenon is attributed to the multiple targets of etravirine in various viruses. As a non-nucleoside inhibitor, etravirine inhibits HIV infection by blocking reverse transcriptase activity by binding in a pocket near the catalytic site [52,53]. Etravirine exhibits high binding energy with both Mpro and RdRp of SARS-CoV-2 and may be used as a potential drug for the treatment of COVID-19 [54]. Moreover, another non-nucleoside inhibitor rilpivirine has been reported to exert antiviral effects by interacting with the RdRp of ZIKV, a key enzyme for the replication of various arboviruses [49]. Similarly, we showed that etravirine acts as a replication inhibitor of viral infection, and there is a strong binding potential between the macrolide skeleton of etravirine and WNV RdRp or alphavirus RdRp (Figure 4 and Figure 5). Of note, we found that, although etravirine markedly suppressed TBEV infection in Huh 7 and HUVEC, two peripheral cell lines, it had little effect on TBEV infection in SY5Y cells, a neuronal cell line (Figure 3B,E,F). This result means its potential for improving the neuroinvasion and pathogenicity of TBEV may be limited. In contrast, an application at the early infection stage before TBEV enters the central nervous system may be beneficial. Moreover, the inhibitory effect of etravirine on TBEV infection of peripheral target cells was not as strong as that on YFV and WNV, supposing the combination of other antiviral drugs may be beneficial in future applications for TBEV treatment.

In vivo, repeated treatment with etravirine significantly inhibited WNV infection, improving the survival rate by approximately 50%, reducing the percentage of weight loss by approximately 8%, and decreasing the viral titer during organ infection. WNV infection causes severe central nervous system diseases, such as encephalitis, which are caused by inflammatory injury. CCL-2 was reported to mediate the accumulation of inflammatory monocytes in the brain, and their differentiation into microglia decreased survival, thus playing a pathogenic role in encephalitis caused by WNV [55]. Blocking IL-10 signaling reportedly promoted survival after a lethal WNV challenge in mice [56]. We also found that the expression of *IL-6*, *IL-10*, *CCL-2,* and *TNF-α* increased in WNV-DMSO but decreased in etravirine-treated cells, which further supports the conclusion that etravirine suppresses WNV infection both in vitro and in vivo.

The increase in disease incidence and persistent CHIKV-induced arthritis-like symptoms has been a considerable burden on public health across the globe. To evaluate the antiviral activity of etravirine in vivo, a CHIKV-induced arthritis mouse model was utilized. The results demonstrated that etravirine treatment effectively ameliorated footpad swelling and reduced muscle viral burdens at 4 dpi (Figure 7B–D). Like many arbovirus damage mechanisms, the inflammatory response is the major cause of arthritis associated with CHIKV infection [55,56,57]. Etravirine effectively decreased the levels of immunomodulatory host factors (*IL-6*, *IL-10*, *CCL-2,* and *TNF-α*) that were triggered by CHIKV infection. It is believed that CHIKV infection has a low fatality rate, but since the large outbreak that occurred in 2005–2006, CHIKV infection has been assumed to have evolved to a more severe form of the disease with the central nervous system (CNS) involvement [58]. With this in mind, we showed that etravirine had a protective effect on a CHIKV lethal mouse model, in which it exhibits a therapeutic antiviral effect (Figure 8).

## 5. Conclusions

This study illustrated the validated anti-arbovirus ability of etravirine in both Huh 7 and SH-SY5Y cells, including WNV, YFV, TBEV, and CHIKV. The drug is involved in the virus replication process, potentially due to binding to the RdRp of arbovirus in molecular docking assay. In WNV infected mice model, etravirine significantly rescued mice from viral infection-induced weight loss, severe neurological symptoms, and death and reduced the viral load and inflammatory cytokines in target tissues. In the arthrophlogosis and lethal mouse models, etravirine also has antiviral effects on CHIKV infection. This make it a promising candidate for clinical application for arbovirus infection.

## Figures and Tables

**Figure 1 pharmaceutics-16-01111-f001:**
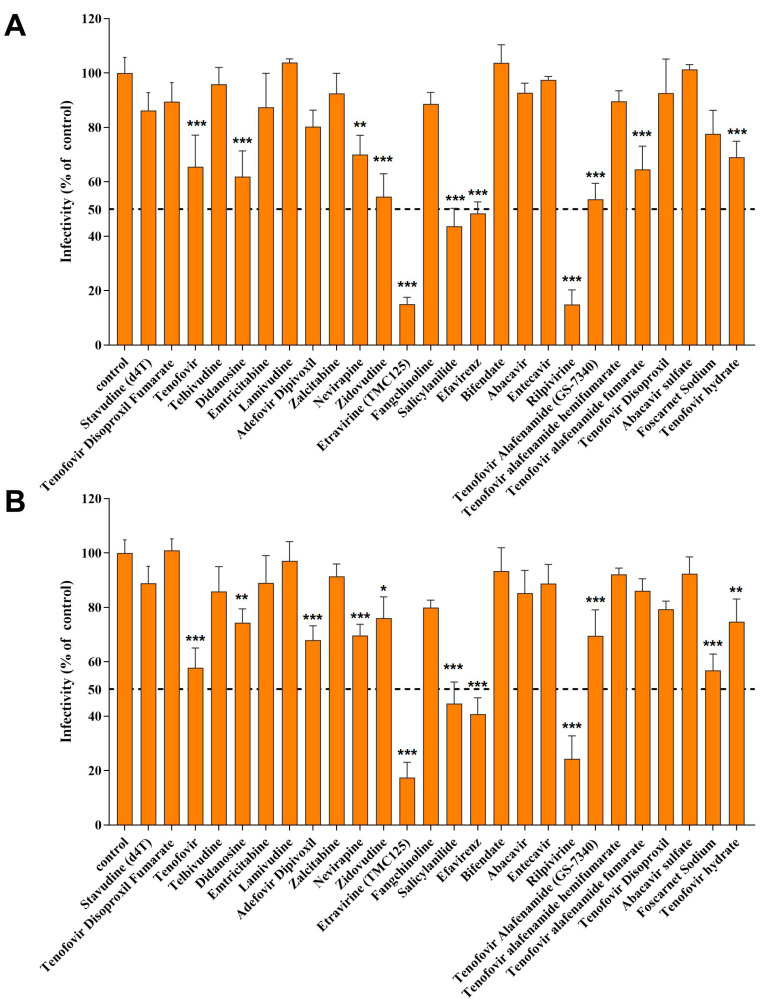
Screening of FDA-approved reverse transcriptase inhibitors for WNV infection. After being incubated with WNV (MOI = 0.1) for 2 h, Huh 7 cells (**A**) and SH-SY5Y cells (**B**) in 96-well plates were washed and treated with FDA-approved reverse transcriptase inhibitors or DMSO (control) at a 25 μM concentration. Twenty-four hours post-infection (24 hpi), cells were fixed and IF assays were performed to detect WNV. The percentage of viral antigen-positive cells was calculated and normalized to DMSO-treated group (control). Compounds with ≥50% inhibitory rate for WNV infectivity (dotted lines) compared with the control group were considered for further investigation. (n = 3) * *p* < 0.05, ** *p* < 0.01, *** *p* < 0.001 compared with DMSO control.

**Figure 2 pharmaceutics-16-01111-f002:**
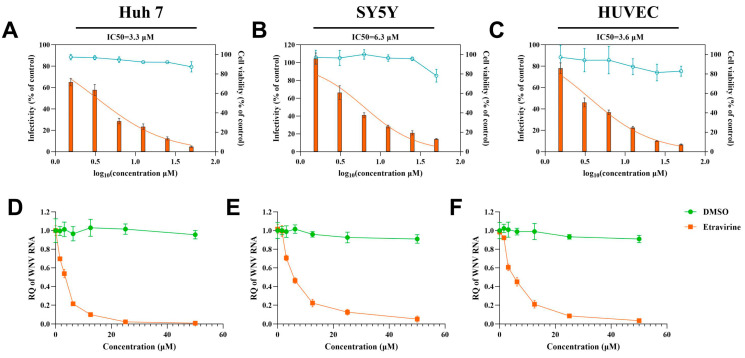
Etravirine inhibits infection of WNV in Huh 7 cells, SH-SY5Y cells, and HUVEC in a dose-dependent manner. After being incubated with WNV (MOI = 0.1) for 2 h, Huh 7 (**A**,**D**), SH-SY5Y (**B**,**E**), and HUVEC (**C**,**F**) were treated with different concentration (50, 25, 12.5, 6.25, 3.125, and 1.625 μM) of etravirine or DMSO. (**A**–**C**) Cells were suffered from IF assays at 24 hpi and the percent of positive cells normalized to the DMSO-treated group (control). Orange histograms represent the relative infection rate. The IC50 values were calculated using the nonlinear regression model (orange lines, the parameters of the models are listed in Appendix A). Drug-treated but uninfected cells were tested by a CCK-8 kit and the percent of cell viability was normalized to the control group. Blue lines represent the relative cell viability; (**D**–**F**) viral RNA copies in cells were quantified 24 hpi by RT-qPCR and results are denoted as the relative quantity (RQ) of WNV RNA genome compared with the control group. (n = 6) The results are representative of three independent experiments.

**Figure 3 pharmaceutics-16-01111-f003:**
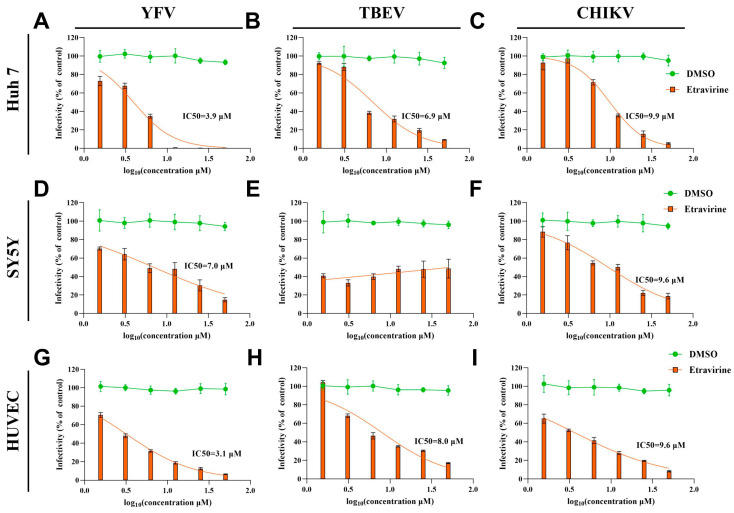
Etravirine exhibits antiviral activity to YFV, TBEV, and CHIKV in multiple cell lines. Cultured Huh 7 (**A**–**C**), SH-SY5Y (**D**–**F**), and HUVEC (**G**–**I**) were infected by YFV (MOI = 0.1) (**A**,**D**,**G**), TBEV (MOI = 0.1) (**B**,**E**,**H**), and CHIKV (MOI = 0.1) for 2 h, followed by treatment of six 2-fold serial dilution (50–1.625 μM) of etravirine or DMSO. Twenty-four hours post infection (24 hpi), IF assays were performed to detect viral infection. The percentage of viral antigen-positive cells was calculated and normalized to the DMSO-treated group (control), and the IC50 values were calculated using the nonlinear regression model (orange lines, the parameters of the models are listed in Appendix A). The results are representative of three independent experiments.

**Figure 4 pharmaceutics-16-01111-f004:**
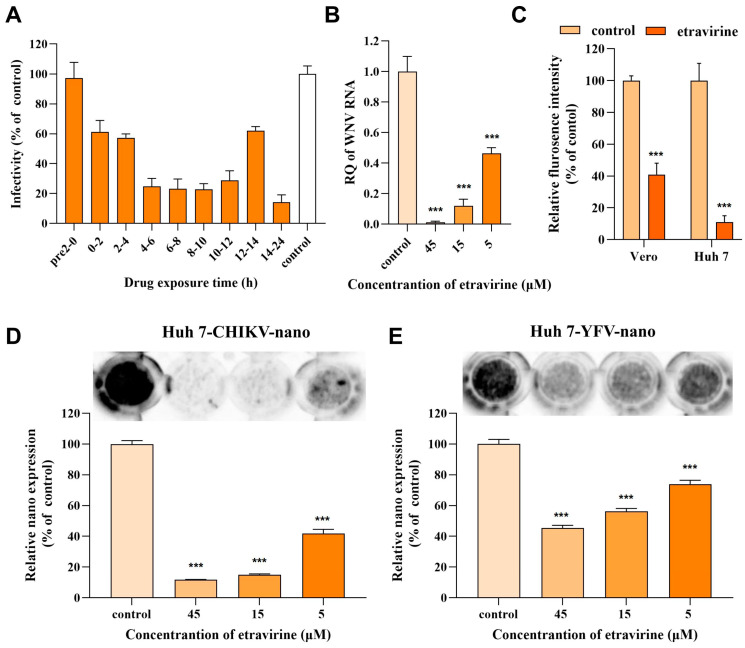
Etravirine exerts its antiviral effect by inhibiting viral replication. (**A**) Time-of–drug-addition assays were performed (see the Methods Section) to determine the action stage of etravirine: at different time point of WNV infection (MOI = 0.1), etravirine (25 μM) or DMSO was added and incubated for 2 h; Huh 7 cells were then washed and cultured to maintain 24 h of infection, followed by IF assays to detect WNV. Relative infectivity of each period was calculated and normalized to the DMSO control. (**B**) A genome transient expression assay was performed to assess WNV replication under treatment: WNV genome was isolated from virion and transfected into Huh 7 cells. Six hours post-transfection (6 hpt), the supernatant was replaced with etravirine containing fresh medium and treated for four hours before the cells were lysed and viral load was analyzed with RT-qPCR. (**C**) After being transfected with CHIKV replicon RNA carrying EGFP reporter gene for 6 h, Vero or Huh 7 cells were treated with 25 μM of etravirine or DMSO. Twenty-four hours later, the fluorescence was imaged. The mean fluorescence intensity CHIKV-LR2006-EGFP was calculated and normalized to the DMSO-treated control. (**D**,**E**) After being transfected with CHIKV (CHIKV-LR2006-nano) or YFV (YFV-17D-nano) replicon RNA carrying NanoLuc reporter gene for 6 h, Huh 7 cells were treated with different concentrations of etravirine or DMSO. Twenty-four hours later, NanoLuc reagent was added and the luminescence was captured. The gray value of each well was analyzed by ImageJ 1.47. The data were normalized to the DMSO-treated group (control). *** *p* < 0.001 compared to DMSO control. The results are representative of three independent experiments.

**Figure 5 pharmaceutics-16-01111-f005:**
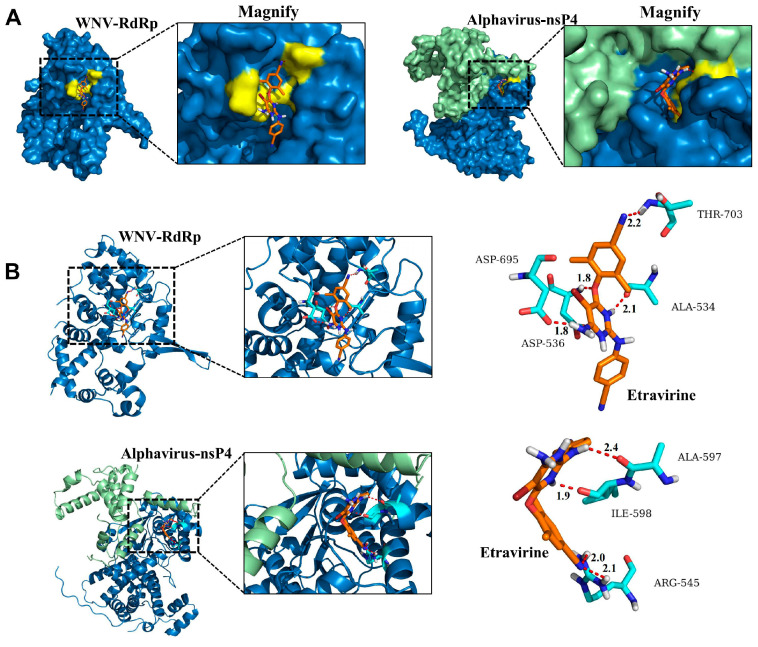
Molecular docking analysis between etravirine and WNV NS5 RdRp or alphavirus nsP4. The binding of etravirine to WNV NS5 RdRp or alphavirus nsP4 was simulated by computer docking, and three-dimensional (3D) ligand interaction maps of RdRp or nsP4 bound with etravirine were generated. (**A**) WNV RdRp (left) is shown as a blue surface, two chains of alphavirus nsP4 (right) are shown as a green or blue surface, respectively, and etravirine is shown as orange sticks. The inset shows magnified boxed areas. Interact residues lining the pocket for etravirine are marked in yellow. (**B**) Target proteins in (**A**) are shown as cartoon models and etravirine was shown as colored sticks. Residues lining the pocket for etravirine (incarnadine sticks) are shown as sky-blue sticks. Residues interacting with etravirine are labeled according to their numbering in WNV NS5 RdRp (PDB: 2HCN) or alphavirus nsP4 (PDB: 7VW5). Dashed lines represent the hydrogen bonds and the numbers represent their distances.

**Figure 6 pharmaceutics-16-01111-f006:**
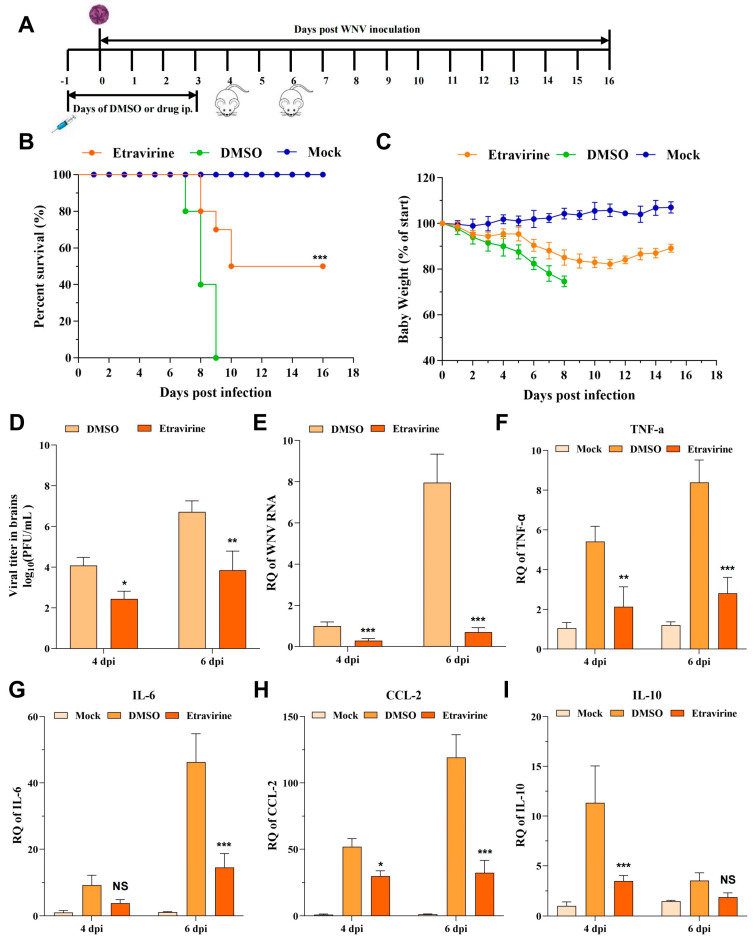
Etravirine protects mice from WNV infection-caused injury and death. (**A**) Scheme of the in vivo experiment. (**B**,**C**) After treatment, the survival (**B**) and percentage of weight change (relative to starting weight) (**C**) were monitored daily for 16 days. (n = 10) (**D**–**I**) Brains harvested at 4 and 6 dpi were analyzed for viral load by plaque assay (**D**) or RT-qPCR (**E**) or for the expression of inflammatory cytokines, like *TNF-α* (**F**), *IL-6* (**G**), *CCL-2* (**H**), and *IL-10* (**I**) by RT-qPCR. Mock group: uninfected and oral administration of DMSO; DMSO group: infected and oral administration of DMSO; etravirine group: infected and oral administration of 40 mg/kg of etravirine. (n = 6). * *p* < 0.05, ** *p* < 0.01, *** *p* < 0.001 compared with the DMSO control. NS, not significant. The results are representative of three independent experiments.

**Figure 7 pharmaceutics-16-01111-f007:**
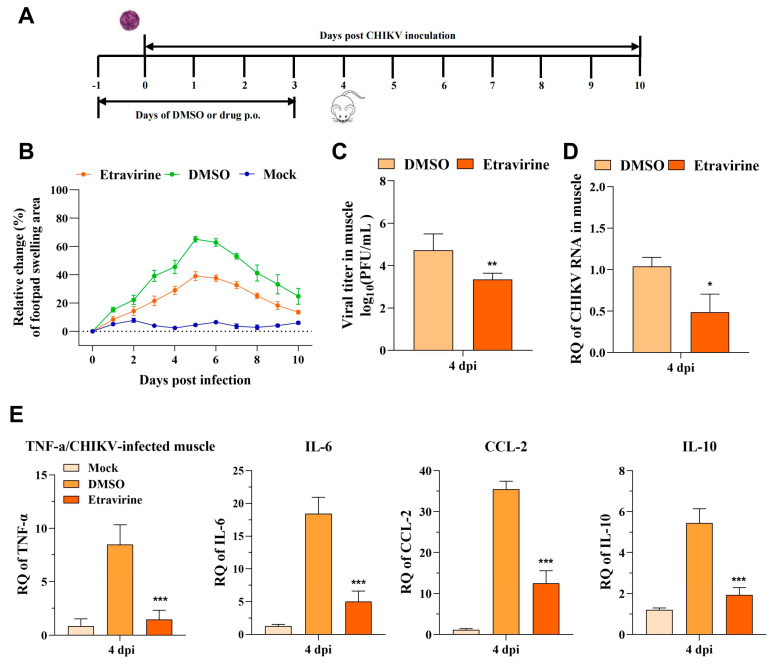
Etravirine protects mice from CHIKV infection-caused injury. (**A**) Scheme of the in vivo experiment. (**B**) Mice were monitored daily for left rear footpad swelling for 10 days (n = 10). (**C**–**E**) The muscle tissue was harvested at 4 dpi and viral load was assessed by a plaque assay (**C**) or RT-qPCR (**D**), and the expression of inflammatory cytokines was evaluated by RT-qPCR (**E**). Mock group: uninfected and oral administration of DMSO; DMSO group: infected and oral administration of DMSO; etravirine group: infected and oral administration of 40 mg/kg of etravirine. * *p* < 0.05. ** *p* < 0.01, *** *p* < 0.001 compared with the DMSO control. NS, not significant. (n = 3). The results are representative of three independent experiments.

**Figure 8 pharmaceutics-16-01111-f008:**
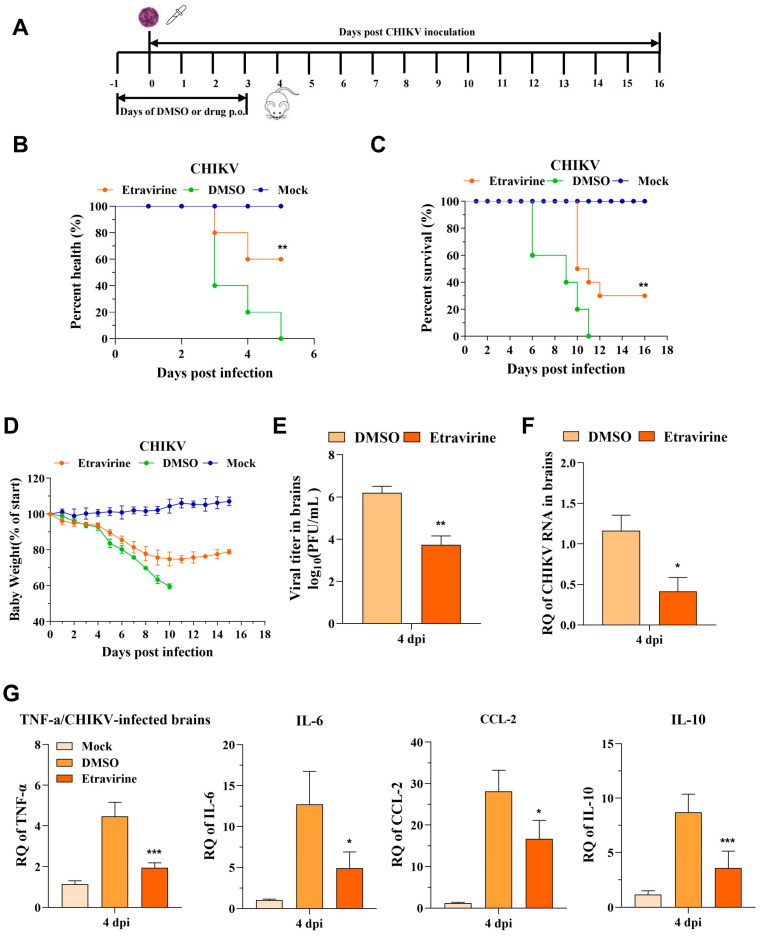
Etravirine protects mice from CHIKV infection-caused injury and death. (**A**) Scheme of the in vivo experiment. (**B**) After treatment, the health index of CHIKV ROSS strain-infected mice was assessed by statistical analysis of the hunched posture, ruffled coat, and mobility activity (n = 10). (**C**,**D**) Survival curves and percentage of weight change (relative to starting weight) after treatment (n = 10). (**E**–**G**) Brains were harvested at 4 dpi and viral load was assessed by a plaque assay (**E**) or RT-qPCR (**F**), and the expression of inflammatory cytokines was tested by RT-qPCR (**G**). Mock group: uninfected and oral administration of DMSO; DMSO group: infected and oral administration of DMSO; etravirine group: infected and oral administration of 40 mg/kg of etravirine. * *p* < 0.05. ** *p* < 0.01, *** *p* < 0.001 compared with the DMSO control. NS, not significant. (n = 3). The results are representative of three independent experiments.

## Data Availability

The data that support the findings of this study are available from the corresponding author upon reasonable request.

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
