# Peer review of "Etravirine Prevents West Nile Virus and Chikungunya Virus Infection Both In Vitro and In Vivo by Inhibiting Viral Replication"

_pharmaceutics, 2024, doi:10.3390/pharmaceutics16091111_

Round 1

Reviewer 1 Report

Comments and Suggestions for Authors

Dear authors,

Thank you for submitting your manuscript on the potential of etravirine as an antiviral agent against arboviruses such as West Nile virus (WNV) and Chikungunya virus (CHIKV). I have thoroughly reviewed the manuscript and have a few serious concerns about the presentation of the materials. Here are my specific comments and suggestions for improvement:

1.       Abstract: The statement in line 16-18 that "Reverse transcriptase inhibitors are important direct antiviral drugs for the treatment of HIV or HBV infection, some of which have been repurposed in studies combating these infections" is confusing. WNV and CHIKV are both RNA viruses and do not have reverse transcriptase. HIV is a retrovirus and HBV is a DNA virus, making their life cycles and requirements very different. I suggest rephrasing it to convey that the study aims to identify new sources of antiviral small molecules, and FDA-approved inhibitors of reverse transcriptase were tested in this work.

2.      Introduction: It would be beneficial to add more information about the Flavivirus family and mention Zika virus (ZIKV) in lines 45-46. This is important because in the next paragraph, you reference #14 a study (Wang B et al. 2018) that discusses the NS5 protein of ZIKV. Additionally, the reference #13 (Keating JA et al. 2013) cited in line 57-58 does not support the statement that RdRp is one of the most promising targets for anti-WNV drug development because it is discussing the role of PK-G in phosphorylation of methyltransferase of NS5. More discussion about the roles of the NS5 protein in interaction with cellular components is needed to make this reference fit the purpose. The introduction is also lacking information about tissue specificity of the viruses, which would explain the design of the experimental tests, selection of cell lines, virus models, and the sensitivity of mouse models to the viruses used in this study. Reference #24 (Brochard T et al. 2023) does not support the statement in line 78-79 that repurposing drugs requires less time and cost than de novo drug discovery. This reference discusses the repurposing of NRTIs to suppress DNA damage by retrotransposons in aging mice. I suggest discussing potential mechanisms by which NRTIs can work in cells infected with RNA flaviviruses to make this reference more relevant. Also, consider including references that discuss successful stories of drug repurposing for antiviral treatments to justify the selected approach. For instance:  Travedi J . et al. 2020 – “Drug Repurposing Approaches to Combating Viral Infections”, J.ClinMed. 9(11):3777

3.      Materials and Methods: In part 2.2 (lines 120-121), you mention an immunofluorescence assay that is described in Section 2.4, and not in the supplemental materials. Please correct this. Furthermore, reference #29 (Liu YG et al. 2020) does not provide information on the "drug inhibition assay" or immunofluorescent staining. Additionally in the part describing Use of GraphPad Prism for IC50 calculations, it would be helpful to specify the models used for curve-fitting to calculate the EC50 values, as the curves presented in Figure 2 and most of Figure 3 do not have all parameters for traditional 4-parametric logistic regression.

4.      Immunofluorescence Assay (Section 2.4): Reference #31 (Qian X et al. 2022) does not contain information about the immunofluorescent assay. The phrase in lines 139-142, stating that cells were incubated with primary antibodies against the virus, which were prepared by immunizing rabbits with formaldehyde-inactivated virus in the laboratory (overnight, 4C), needs to be rephrased as it currently does not make sense. Additionally, reference #32 (Hr Y et al. 2023) mentions that antibodies against CHIKV E1 were purchased, not produced in the laboratory. Clarification is needed. It would be helpful to provide a list of antibodies and their sources used in the immunostaining tests.

 Criteria for thresholding "infected" and "non-infected" cells should be clearly described.

Also the infection rates should be presented, especially considering the low MOI used and potential low total virus infection after media removal.

5.      Real-time qPCR (Section 2.6): The description of experimental replicates needs to be added to the methods, as well as for every graph showing statistical significance.

6.      Replication Determination (Section 2.7): The sources of each viral construct presented in this section are not clearly explained and are confusing. Reference 31 (Qian X et al. 2022) does not describe the generation of YFV-Nano Luc plasmid, but rather references the work done by Patkar CG et al. in 2009. There is also no information about the YFV-GFP used in the test or it is not clearly presented.

7.       In vivo experiment (Section 2.8): There is no mentioning of the plaque assay design or a reference to the published protocol, despite the results of the virus load in brain tissues being shown in Figures 6, 7, and 8. The description of the animal treatment is not clear, and it would be better to describe the lethal and subcutaneous models for CHIKV tests separately. Clear information should be provided in the methods and duplicated information in the figure descriptions should be reduced.

8.      Results: In Section 3.1, it would be helpful to comment on the observed infection rate before normalization to control after using an MOI of 0.1 for such a short time, washing the virus, and incubating for only 24 hours after infection. If the observed infection rate is too low, it could explain the strong but non-specific effect of the tested NNRTIs. The discussion about studies done with ZIKV in lines 250-253 should be moved from the Results section to the Discussion section.

9.      Section 3.2: Information about arboviruses (lines 264-265) should be discussed in the explanation of the experimental design in the introduction or in the discussion section. The models used for IC50 calculation need better explanation, as there are not enough parameters for traditional 4-parametric logistic regression.

10.   There is an error in the description of results in line 281, where "less toxic" is mistakenly referred to as "more toxic."

11.     There is inaccurate interpretation and description of results:

  • Line 281: The phrase "it was slightly less toxic to SY5Y cells, with a decrease in cell viability of approximately 21.8%" incorrectly refers to a more toxic effect as "less toxic." Please correct this in your revision.
  • Line 285-286: The phrase "we found that the antiviral effect of etravirine on SY5Y cells was also weaker than that on SY5Y cells" has a clear error. Please revise it.
  • Line 287: The use of the term "speculation" in relation to the results shown in Figure 1 is incorrect. Please provide an accurate description in your revision.

12.    Rephrasing of figure legends:

  • Lines 290-299: The figure legends for Figure 2 require rephrasing to enhance clarity. Specifically, the sub-labels for graph A, B, and C should precede the corresponding descriptions, distinguishing the orange histograms as representing percent of positive cells and blue lines as representing percent of cell viability.
  • The graphs in Figure 4 are too small, making it challenging to read the information on the axes. Consider organizing the figure differently or separating it into two figures, one for A, B, and C, and another for D and E, to improve clarity.
  • Figure 5: The phrase "was shown as colored sticks" in the description of Figure 5 should be revised to "colored lines." Additionally, since all parts of the image are colored, specifying "orange color lines" may be more appropriate.

13.   Clarification of the molecular docking method is needed:

  • Lines 390-411: The section on molecular docking should be separated into a subsection and provide more information about the analysis conducted and the criteria used. Additionally, please ensure that the method for molecular docking is also adequately described in the Methods section of the manuscript.
  • Lines 390-396: The content from this fragment appears to fit better in the discussion section rather than the results section. Please move it accordingly.
  • Line 408-410: The calculations performed to demonstrate hydrogen bonds between etravirine and the alphavirus RdRp residue need to be described. Please provide these details in your revision.

14.   For in vivo studies:

  • The clarification of the chosen dose of 40mg/kg of etravirine is needed with the reference to the pharmacokinetics/pharmacodynamics (PK/PD) of etravirine in mice and  should be included in the Methods or Discussion section. Additionally, please avoid abbreviating the word "days" to "d" in the text.
  • Figures 6, 7, and 8 contain repetitive information from the Methods section that does not add critical information for understanding the figures. Please revise the descriptions to provide more relevant details. Additionally, considering the small fonts and size of the graphs, it is challenging to read the labels on the axes. Please increase the font size for better readability. It would also be helpful to indicate the number of replicates used in each test for each sample.

15.   Reference #25 (Namasivayam V. et al 2019) cannot be used to illustrate the "broad-spectrum antiviral properties" of etravirine against SARS-CoV-2 and HIV, as stated in line 552-554. It is talking about development of NNRTI etravirine targeting specifically HIV-1. Please revise.

16.   Line 559-560: The phrase "Currently, etravirine also exhibits high binding energy with the RdRp of SARS-CoV-2 against dominant omic variants" requires rephrasing for accuracy.

17.    Lack of accuracy and supporting evidence in the conclusions:

  • The conclusions drawn in the manuscript are not sufficiently accurate and are not based on the experimental results presented. Please ensure that your conclusions are based on the evidence provided in the study.
  • It would be beneficial to include a discussion on the disadvantages of NNRTIs, including etravirine, such as the fast rate of developing resistance, drug interactions, and issues with non-specific types of inhibitors.
  • Line 562: The conclusion that "four potential hydrogen bonds were formed" needs to be explained, as the results do not provide information on how these bonds were calculated or confirmed.
  • The discussion on the observed results for TBEV in line 566, specifically the phrase "an application in the early infection stage before TBEV enters the central nervous system may be beneficial," needs to be better explained and accurately linked to the presented results.
  • Lines 567-570: The phrase "Moreover, the inhibitory effect of etravirine on TBEV infection of peripheral target cells (hepatocytes and vascular endothelial cells) was not as strong as that on TBEV infection of YFV and WNV, suggesting that the combination of other antiviral drugs is needed for future applications" should be rephrased for accuracy.
  • Line 581: The conclusion that "etravirine is resistant to WNV infection both in vitro and in vivo" is incorrect and needs to be revised.
  • Line 597: The formulation of the conclusion needs to be more accurate, such as "we showed that etravirine is potentially involved in the virus replication process and has the ability to inhibit the arbovirus by binding to the RdRp" since the binding was not experimentally shown but suggested or calculated using molecular docking.

Please consider these revisions and provide a more accurate and scientifically supported manuscript. We look forward to reviewing the revised version of your study.

Sincerely

Comments on the Quality of English Language

Those are only few of multiple Grammer and Language problems. Few more are also listed in report to authors.:

Lines 14-15: “ Chikungunya virus (CHIKV) is alarming, as is the treatment of global public health”. The revised sentence would read as, Chikungunya virus (CHIKV) poses a threat to global public health…

Lines 22-26: The sentence :” A time-of-drug-addition assay based on an…” is very long and busy with information and need to be re-phrased to make it clear for understanding. needs to be rephrased to improve clarity and readability.

Line 66-67: The sentence "A characteristic feature of CHIKV disease is recurring musculoskeletal disease primarily affecting the..." should be rephrased for better clarity. One possible rephrasing could be, "One notable characteristic of CHIKV disease is the recurrence of musculoskeletal symptoms, particularly affecting the..."

Line 72-73: The sentence "Similar to WNV, some compounds targeting RdRp, such as favipiravir T-705 and sofosbuvir, have been reported to inhibit the replication cycles of CHIKV and other alphaviruses." needs to be revised as the phrase "similar to WNV" is not clear .

Line 233: The titles of section 3.1 "Screening of FDA-approved reverse transcriptase inhibitors for WNV infection"  and in Line 256 The title of Figure 1 "Screening of FDA-approved reverse transcriptase inhibitors on WNV infection" – should be corrected

Line 305: To improve clarity, the phrase "were treated with different concentrations of etravirine, and viral infectivity was detected"  by re-phrasing “..and viral infectivity was assessed using methods..."

Lines 459-462: The phrase "If etravirine relieves arthrophlogosis in mice, we infected the mice with the CHIKV LR2006 OPY1 strain via subcutaneous injection in the left rearfoot with or without etravirine (40 mg/kg) treatment for 5 d (Figure 7A)." needs to be rephrased for better flow.

Lines 527-528: The phrase "This study focused on FDA-approved reverse transcriptase inhibitors to detect anti-WNV candidates in vitro and in vivo" should be rephrased. One possible revision could be, "The primary objective of this study was to evaluate FDA-approved reverse transcriptase inhibitors as potential candidates with anti-WNV activity, both in vitro and in vivo."

  • Lines 567-570: The phrase "Moreover, the inhibitory effect of etravirine on TBEV infection of peripheral target cells (hepatocytes and vascular endothelial cells) was not as strong as that on TBEV infection of YFV and WNV, suggesting that the combination of other antiviral drugs is needed for future applications" should be rephrased for accuracy.
  • Line 581: The conclusion that "etravirine is resistant to WNV infection both in vitro and in vivo" is incorrect and needs to be revised.

Author Response

Comments 1: Abstract: The statement in line 16-18 that "Reverse transcriptase inhibitors are important direct antiviral drugs for the treatment of HIV or HBV infection, some of which have been repurposed in studies combating these infections" is confusing. WNV and CHIKV are both RNA viruses and do not have reverse transcriptase. HIV is a retrovirus and HBV is a DNA virus, making their life cycles and requirements very different. I suggest rephrasing it to convey that the study aims to identify new sources of antiviral small molecules, and FDA-approved inhibitors of reverse transcriptase were tested in this work.

Response 1: Thank you very much for your suggestion. We revised related statement in the “Abstract” part of the revised manuscript (Line 15-17, Page 1).

Comments 2: Introduction: It would be beneficial to add more information about the Flavivirus family and mention Zika virus (ZIKV) in lines 45-46. This is important because in the next paragraph, you reference #14 a study (Wang B et al. 2018) that discusses the NS5 protein of ZIKV. Additionally, the reference #13 (Keating JA et al. 2013) cited in line 57-58 does not support the statement that RdRp is one of the most promising targets for anti-WNV drug development because it is discussing the role of PK-G in phosphorylation of methyltransferase of NS5. More discussion about the roles of the NS5 protein in interaction with cellular components is needed to make this reference fit the purpose. The introduction is also lacking information about tissue specificity of the viruses, which would explain the design of the experimental tests, selection of cell lines, virus models, and the sensitivity of mouse models to the viruses used in this study. Reference #24 (Brochard T et al. 2023) does not support the statement in line 78-79 that repurposing drugs requires less time and cost than de novo drug discovery. This reference discusses the repurposing of NRTIs to suppress DNA damage by retrotransposons in aging mice. I suggest discussing potential mechanisms by which NRTIs can work in cells infected with RNA flaviviruses to make this reference more relevant. Also, consider including references that discuss successful stories of drug repurposing for antiviral treatments to justify the selected approach. For instance:  Travedi J . et al. 2020 – “Drug Repurposing Approaches to Combating Viral Infections”, J.ClinMed. 9(11):3777

Response 2: Thank you for your kind suggestions. we corrected related statement as follow:

(1)We added the information about the Flavivirus family and Zika virus in the “Introduction” part of the revised manuscript (Line 51--54, Page 2);

(2)We also instead of reference#13 (Keating JA et al. 2013) with a relevant reference about NS5 is a target for antiviral drug reference#24-26 (Malet H et al. 2008; García-Zarandieta M et al. 2023; Wang B et al. 2018, Line 74--75, Page 2);

(3)Add discussion about the roles of the NS5 protein in interaction with cellular components is a good idea, we administrated related statement in the “Introduction” part of the revised manuscript (Line 70--73, Page 2);

(4)We also mention invasion routes and replication sites of WNV to explain the design of the experimental tests, selection of cell lines in this study (Line 56--58, Page 2);

(5)We replace reference#24 (Brochard T et al. 2023) with a relevant reference about repurposing drugs (reference#31-34; Pushpakom S. et al. 2019, Trivedi J. et al. 2020, Li X. et al. 2021, Zeng S. et al. 2019), and add successful cases of drug repurposing for antiviral treatments (reference#35-36, Zhang J. L. et al. 2021; Reina J. et al. 2022) (Line 79--82, Page 2).  

Comments 3: Materials and Methods: In part 2.2 (lines 120-121), you mention an immunofluorescence assay that is described in Section 2.4, and not in the supplemental materials. Please correct this. Furthermore, reference #29 (Liu YG et al. 2020) does not provide information on the "drug inhibition assay" or immunofluorescent staining. Additionally in the part describing Use of GraphPad Prism for IC50 calculations, it would be helpful to specify the models used for curve-fitting to calculate the EC50 values, as the curves presented in Figure 2 and most of Figure 3 do not have all parameters for traditional 4-parametric logistic regression.

Response 3: Thank you for your reminding. The instruction of immunofluorescence assay was indeed described in part 2.4 and provided a reference that have a description of IF assay marked in yellow-highlight in the revised manuscript. (Line 133--134, Page 3). Furthermore, a new supplementary table (“Table S4”) containing the parameters of the nonlinear regression models was provided in the revised files and stated in the figure legend of Figure 2 and Figure 3.

Comments 4: Immunofluorescence Assay (Section 2.4): Reference #31 (Qian X et al. 2022) does not contain information about the immunofluorescent assay. The phrase in lines 139-142, stating that cells were incubated with primary antibodies against the virus, which were prepared by immunizing rabbits with formaldehyde-inactivated virus in the laboratory (overnight, 4C), needs to be rephrased as it currently does not make sense. Additionally, reference #32 (Hr Y et al. 2023) mentions that antibodies against CHIKV E1 were purchased, not produced in the laboratory. Clarification is needed. It would be helpful to provide a list of antibodies and their sources used in the immunostaining tests. Criteria for thresholding "infected" and "non-infected" cells should be clearly described. Also the infection rates should be presented, especially considering the low MOI used and potential low total virus infection after media removal.

Response 4: Thank you for your viable suggestions. we corrected related statement as follow:

(1)As the reviewer reminded, we checked the reference #31 (Qian X et al. 2022). Indeed, they have described the immunofluorescent assay in the “Supplementary Material” part. We also replaced it with reference #40 (He Y et al. 2023) which has a description of immunofluorescent assay in Materials and methods. (Line 134, Page 3).

(2)As suggested by the reviewer, we simplified the statement about primary antibodies and provided a supplementary table (“Table S3”) containing the information of the antibodies in the revised files (the error in CHIKV antibody was corrected).

(3)The criteria for thresholding "infected" cells was stated in the section 2.4 as “The infected (with fluorescence intensity more than 5000) and total cells were counted using Gen5 3.10” (Line 141--142, Page 3).

(4) Different viruses show different infectivity to the same cell, and the same virus exhibits different infectivity to different cells. In order to compare the effects of drugs on different viruses in different cells, we characterized the infectivity by using the normalized infection rate. In terms of infection rate, initial infection with a MOI of 0.1 typically results in about 30% infection rate, at which overlap of green fluorescence between adjacent infected cells can be avoided. After 2 h of incubation, most of the virions are binding to the surface of cells, thus media removal always has little impact on infection.

Comments 5: Real-time qPCR (Section 2.6): The description of experimental replicates needs to be added to the methods, as well as for every graph showing statistical significance.

Response 5: Thank you for your kind suggestions. we added the statement about experimental replicates in the “Statistical analysis” part (Line 233-234, Page 5) and the figure legend in revised manuscript.

Comments 6: Replication Determination (Section 2.7): The sources of each viral construct presented in this section are not clearly explained and are confusing. Reference 31 (Qian X et al. 2022) does not describe the generation of YFV-Nano Luc plasmid, but rather references the work done by Patkar CG et al. in 2009. There is also no information about the YFV-GFP used in the test or it is not clearly presented.

Response 6: Thank you for your valuable and thoughtful comments. We further described in this section about the generation of CHIKV-LR2006-EGFP, CHIKV-LR2006-NanoLuc, YFV-17D-NanoLuc plasmid (details in Figure S2).

Comments 7: In vivo experiment (Section 2.8): There is no mentioning of the plaque assay design or a reference to the published protocol, despite the results of the virus load in brain tissues being shown in Figures 6, 7, and 8. The description of the animal treatment is not clear, and it would be better to describe the lethal and subcutaneous models for CHIKV tests separately. Clear information should be provided in the methods and duplicated information in the figure descriptions should be reduced.

Response 7: Thank you for your suggestions. We corrected related statement as follow:

(1)References of the plaque assay was added (Line 214, Page 5; reference#42 and 43, Tang W D et al. 2023 and Qian X et al. 2022).

(2)We are very sorry for our unclear description about the animal treatment. As suggested, we rephrased the two models by separately described. (Line 216-227, Page 5)

(3) As suggested, we carefully checked the whole paper to reduced unclear and duplicated information.

Comments 8: Results: In Section 3.1, it would be helpful to comment on the observed infection rate before normalization to control after using an MOI of 0.1 for such a short time, washing the virus, and incubating for only 24 hours after infection. If the observed infection rate is too low, it could explain the strong but non-specific effect of the tested NNRTIs. The discussion about studies done with ZIKV in lines 250-253 should be moved from the Results section to the Discussion section.

Response 8: Thank you for your valuable and thoughtful comments. We corrected related statement as follow:

(1)We explained this question in above response. Initial infection with a MOI of 0.1 typically results in about 30% infection rate, and this may not affect the action of tested NNRTIs. In Figure 1, some NNRTIs, such as tenofovir (including its derivant), didanosine, and zidovudine, also show antiviral ability with an efficiency less than 50%. On the one hand, those small molecular may really exhibited anti-WNV activity, while their IC50 may be higher than etravirine and ripivirine. On the other hand, the targets of those compounds may not be the key factor of WNV infection or there is a complementary mechanism to the target. This has been discussed in the revised paper (Line 562-568, Page 16).

(2) In fact, the introduction of rilpivirine inhibits ZIKV infection and replication is explain why we focused on etravirine. Following your feasible suggestion, the reference of rilpivirine was moved from the Results section to the Discussion section.  (Line 555-556, Page 16).

Comments 9: Section 3.2: Information about arboviruses (lines 264-265) should be discussed in the explanation of the experimental design in the introduction or in the discussion section. The models used for IC50 calculation need better explanation, as there are not enough parameters for traditional 4-parametric logistic regression.

Response 9: Thank you for the suggestions. We corrected related statement as follow:

(1)we described the information arboviruses in the introduction part in the revised manuscript (Line 57-58, Page 2)

(2) As suggested, parameters of the nonlinear regression models was provided in the Table S4 of the revised files.

Comments 10: There is an error in the description of results in line 281, where "less toxic" is mistakenly referred to as "more toxic."

Response 10: Thank you for the comment. We are very sorry for the mistake, and it was corrected in the revised manuscript (Line 286-289, Page 7).

Comments 11: There is inaccurate interpretation and description of results:

  • Line 281: The phrase "it was slightly less toxic to SY5Y cells, with a decrease in cell viability of approximately 21.8%" incorrectly refers to a more toxic effect as "less toxic." Please correct this in your revision.
  • Line 285-286: The phrase "we found that the antiviral effect of etravirine on SY5Y cells was also weaker than that on SY5Y cells" has a clear error. Please revise it.
  • Line 287: The use of the term "speculation" in relation to the results shown in Figure 1 is incorrect. Please provide an accurate description in your revision.

Response 11: Thank you for the suggestions. As suggested, the errors or inaccurate interpretation and description of results were corrected (Line 286-289, Page 7; Line 291-294, Page 7).

Comments 12: Rephrasing of figure legends:

  • Lines 290-299: The figure legends for Figure 2 require rephrasing to enhance clarity. Specifically, the sub-labels for graph A, B, and C should precede the corresponding descriptions, distinguishing the orange histograms as representing percent of positive cells and blue lines as representing percent of cell viability.
  • The graphs in Figure 4 are too small, making it challenging to read the information on the axes. Consider organizing the figure differently or separating it into two figures, one for A, B, and C, and another for D and E, to improve clarity.
  • Figure 5: The phrase "was shown as colored sticks" in the description of Figure 5 should be revised to "colored lines." Additionally, since all parts of the image are colored, specifying "orange color lines" may be more appropriate.

Response 12: Thank you for the suggestions. We corrected related statement as follow:

(1)As suggested, the figure legend for Figure 2 was carefully rephrased to enhance clarity.(Line 296-306, Page 7).

(2)The Figure 4 was reorganized for better readability.

(3)The phrase in Figure 5 was revised as suggested.

Comments 13: Clarification of the molecular docking method is needed:

  • Lines 390-411: The section on molecular docking should be separated into a subsection and provide more information about the analysis conducted and the criteria used. Additionally, please ensure that the method for molecular docking is also adequately described in the Methods section of the manuscript.
  • Lines 390-396: The content from this fragment appears to fit better in the discussion section rather than the results section. Please move it accordingly.
  • Line 408-410: The calculations performed to demonstrate hydrogen bonds between etravirine and the alphavirus RdRp residue need to be described. Please provide these details in your revision.

Response 13: Thank you for the suggestions. We corrected related statement as follow:

(1)As reminded, the result of the molecular docking was separated into a subsection (Line 402-421, Page 10-11) and the information of molecular docking was added into the “Materials and Methods” part (2.5, Line 145-156, Page 3-4).

(2)As suggested, we moved the statement to the “Discussion” part (Line 594-598, Page 17).

(3)According to the advice, the distance of the hydrogen bonds between etravirine and the RdRps were calculated using Pymol software, labeled on Figure 5B, and stated in the Methods (Line 145-156, Page 3-4) and Results sections (Line 410-413, Page 10-11; Line 417-419, Page 11).

Comments 14: For in vivo studies:

  • The clarification of the chosen dose of 40mg/kg of etravirine is needed with the reference to the pharmacokinetics/pharmacodynamics (PK/PD) of etravirine in mice and should be included in the Methods or Discussion section. Additionally, please avoid abbreviating the word "days" to "d" in the text.
  • Figures 6, 7, and 8 contain repetitive information from the Methods section that does not add critical information for understanding the figures. Please revise the descriptions to provide more relevant details. Additionally, considering the small fonts and size of the graphs, it is challenging to read the labels on the axes. Please increase the font size for better readability. It would also be helpful to indicate the number of replicates used in each test for each sample.

Response 14: Thank you for the suggestions. We corrected related statement as follow:

(1)As reminded, the reasons for the dose selection are explained in the Methods section (Line 207-208, Page 5). Besides, we checked the whole text and corrected the words.

(2)The figures Figure 6, 7, 8 and their legends were revised as suggested, and the number of replicates were added.

Comments 15: Reference #25 (Namasivayam V. et al 2019) cannot be used to illustrate the "broad-spectrum antiviral properties" of etravirine against SARS-CoV-2 and HIV, as stated in line 552-554. It is talking about development of NNRTI etravirine targeting specifically HIV-1. Please revise.

Response 15: Thank you for the kind reminding. We corrected the reference (Line 586-589, Page 17).

Comments 16: Line 559-560: The phrase "Currently, etravirine also exhibits high binding energy with the RdRp of SARS-CoV-2 against dominant omic variants" requires rephrasing for accuracy.

Response 16: Thank you for the suggestion. We revised the statement as the suggested (Line 594-596, Page 17).

Comments 17: In Lack of accuracy and supporting evidence in the conclusions:

  • The conclusions drawn in the manuscript are not sufficiently accurate and are not based on the experimental results presented. Please ensure that your conclusions are based on the evidence provided in the study.
  • It would be beneficial to include a discussion on the disadvantages of NNRTIs, including etravirine, such as the fast rate of developing resistance, drug interactions, and issues with non-specific types of inhibitors.
  • Line 562: The conclusion that "four potential hydrogen bonds were formed" needs to be explained, as the results do not provide information on how these bonds were calculated or confirmed.
  • The discussion on the observed results for TBEV in line 566, specifically the phrase "an application in the early infection stage before TBEV enters the central nervous system may be beneficial," needs to be better explained and accurately linked to the presented results.
  • Lines 567-570: The phrase "Moreover, the inhibitory effect of etravirine on TBEV infection of peripheral target cells (hepatocytes and vascular endothelial cells) was not as strong as that on TBEV infection of YFV and WNV, suggesting that the combination of other antiviral drugs is needed for future applications" should be rephrased for accuracy.
  • Line 581: The conclusion that "etravirine is resistant to WNV infection both in vitro and in vivo" is incorrect and needs to be revised.
  • Line 597: The formulation of the conclusion needs to be more accurate, such as “we showed that etravirine is potentially involved in the virus replication process and has the ability to inhibit the arbovirus by binding to the RdRp” since the binding was not experimentally shown but suggested or calculated using molecular docking.

Response 17: Thank you for the kind suggestions. We corrected related statement as follow:

(1)We are very sorry for our unclear description, and the conclusion was totally rephrased based on the experimental results to ensure accuracy.

(2)As advised by the reviewer, we discussed the disadvantages of NNRTIs in the revised paper (Line 569-573, Page 16).

(3)The inaccurate statement was revised as reminded (Line 598-601, Page 17).

(4)We rephrased the discussion on the treatment of TBEV accordingly (Line 601-605, Page 17).

(5)The inaccurate statement was revised as reminded (Line 606-609, Page 17).

(6)As suggested, we corrected the statement in the revised manuscript (Line 619-620, Page 17).

(7)We reorganized the conclusion as the reviewer suggested (Line 636-644, Page 18).

Comments 18: Comments on the Quality of English Language:

Those are only few of multiple Grammar and Language problems. Few more are also listed in report to authors.:

  • Lines 14-15: “ Chikungunya virus (CHIKV) is alarming, as is the treatment of global public health”. The revised sentence would read as, Chikungunya virus (CHIKV) poses a threat to global public health…
  • Lines 22-26: The sentence :” A time-of-drug-addition assay based on an…” is very long and busy with information and need to be re-phrased to make it clear for understanding. needs to be rephrased to improve clarity and readability.
  • Line 66-67: The sentence "A characteristic feature of CHIKV disease is recurring musculoskeletal disease primarily affecting the..." should be rephrased for better clarity. One possible rephrasing could be, "One notable characteristic of CHIKV disease is the recurrence of musculoskeletal symptoms, particularly affecting the..."
  • Line 72-73: The sentence "Similar to WNV, some compounds targeting RdRp, such as favipiravir T-705 and sofosbuvir, have been reported to inhibit the replication cycles of CHIKV and other alphaviruses." needs to be revised as the phrase "similar to WNV" is not clear .
  • Line 233: The titles of section 3.1 "Screening of FDA-approved reverse transcriptase inhibitors for WNV infection"  and in Line 256 The title of Figure 1 "Screening of FDA-approved reverse transcriptase inhibitors on WNV infection" – should be corrected
  • Line 305: To improve clarity, the phrase "were treated with different concentrations of etravirine, and viral infectivity was detected"  by re-phrasing “..and viral infectivity was assessed using methods..."
  • Lines 459-462: The phrase "If etravirine relieves arthrophlogosis in mice, we infected the mice with the CHIKV LR2006 OPY1 strain via subcutaneous injection in the left rearfoot with or without etravirine (40 mg/kg) treatment for 5 d (Figure 7A)." needs to be rephrased for better flow.
  • Lines 527-528: The phrase "This study focused on FDA-approved reverse transcriptase inhibitors to detect anti-WNV candidates in vitro and in vivo" should be rephrased. One possible revision could be, "The primary objective of this study was to evaluate FDA-approved reverse transcriptase inhibitors as potential candidates with anti-WNV activity, both in vitro and in vivo."
  • Lines 567-570: The phrase "Moreover, the inhibitory effect of etravirine on TBEV infection of peripheral target cells (hepatocytes and vascular endothelial cells) was not as strong as that on TBEV infection of YFV and WNV, suggesting that the combination of other antiviral drugs is needed for future applications" should be rephrased for accuracy.
  • Line 581: The conclusion that "etravirine is resistant to WNV infection both in vitro and in vivo" is incorrect and needs to be revised.

Response 18: Thank you for the suggestions. We corrected all the problems above according to the thoughtful comments of the reviewer. Additionally, we carefully checked the entire paper and corrected other grammar and language issues.

Reviewer 2 Report

Comments and Suggestions for Authors

Arboviruses has been distributed all over the world and those infectious diseases are recognized as public health threats. Although vaccines are available against some of the arboviruses, no specific antiviral drugs are licensed. Drug reposition is currently considered as an alternative tool to develop antiviral drugs. Authors screened FDA-approved reverse transcriptase inhibitor as anti arbovirus drugs and found that Etravirine inhibited West Nile virus and Chukungunya virus in vitro as well as in vivo. Those results indicated Etravirine is considered as a quite promising anti arbovirus drug candidate. The manuscript is well prepared. Several points were raised.

1.       In the Introduction, please provide worldwide distribution of the viruses addressed in the study.

2.       In the Introduction, please describe on the A226V mutation resulting in spread through the world.

3.       There are not indicators for significant statistical differences in the some of the figure. For example, Were there no significant differences in the figure 1? Please clarify.

4.       Line 286; SY5Y should read as Huh7.

5.       It is encouraged to show the simple structure of the replicon utilized in the study, because those cannot be found on the references.

6.       Have authors examined RdRp mutant to see if the Etravirine affect RNA replication and/or translation?

7.       Figure 6: Can author provide %protection from disease?

Author Response

Comments 1: In the Introduction, please provide worldwide distribution of the viruses addressed in the study.

Response 1: Thank you for the suggestion.As suggested, the worldwide distribution of the viruses addressed in the study was added in the “Introduction” part (Line 40-46, Page 1-2).

Comments 2: In the Introduction, please describe on the A226V mutation resulting in spread through the world.

Response 2: Thank you for the kind reminding. The advice of the reviewer is meaningful. The A226V mutation in E1 glycoprotein facilitates cholesterol-dependent entry and has been shown to increase replication and transmission in mosquitoes, which may lead to increased prevalence of  CHIKV. This description  was inserted in the “Introduction” part. (Line 46-49, Page 2).

Comments 3: There are not indicators for significant statistical differences in the some of the figure. For example, Were there no significant differences in the figure 1? Please clarify.

Response 3: Thank you for the kind reminding, the missing statistical indicators was marked in the revised manuscript.

Comments 4: Line 286; SY5Y should read as Huh7.

Response 4: Thank you for the suggestion, we modified the statement in the revised manuscript (Line 291-294, Page 7).

Comments 5: It is encouraged to show the simple structure of the replicon utilized in the study, because those cannot be found on the references.

Response 5: Thank you for the valuable and thoughtful suggestion, As suggested, we provided schematic diagrams of the replicons’ structure in the supplementary material (“Figure S2”).

Comments 6: Have authors examined RdRp mutant to see if the Etravirine affect RNA replication and/or translation?

Response 6: Thank you for the suggestion. Examination on the effect of RdRp mutant can provide molecular basis for studying the replication of arboviruses and the antiviral mechanism of drugs. However, it is difficult to rescue arboviruses using genetic materials, which limits the study of RdRp mutations in authentic viruses. Therefore, we tried to introduce mutations into the replicons. Unfortunately, the expression of reporter gene was almost undetectable after transfection of the replicon (CHIKV-LR2006-EGFP) RNA with the RdRp mutations (A597R, R545N). Since the reporter gene expression of the original replicon was relatively low, it is possible that the introduced mutations further affect the function of the replicon.

Comments 7: Figure 6: Can author provide %protection from disease?

Response 7: Thank you very much for the suggestion. We provided the percent of protection as suggested and adjusted relevant description (Line 444-447, Page 12).

Round 2

Reviewer 1 Report

Comments and Suggestions for Authors

Dear Authors,

Thank you for addressing all the suggestions and issues so thoroughly. I have one ethical concern that the editor did not previously raise: the mice were left paralyzed for a week before they succumbed to death. I would appreciate it if you could include a description in the methods section to address how the animals' suffering was minimized during this period.

Your manuscript offers valuable observations that will undoubtedly interest the scientific community served by this journal. 

Best regards

Author Response

Comments 1: Thank you for addressing all the suggestions and issues so thoroughly. I have one ethical concern that the editor did not previously raise: the mice were left paralyzed for a week before they succumbed to death. I would appreciate it if you could include a description in the methods section to address how the animals' suffering was minimized during this period.

Response 1: Thank you very much for this valuable suggestion. We are so sorry for result description caused misunderstanding about disable and paralysis. According to observing, under the “disable and paralysis” status, the viral-infected mice could drinking and eating by themselves, although they exhibiting impaired mobility. We also consulted the experts, they suggested that mobility impaired are probably a more accurate description of the infected mice. As yours helpful suggested, we reedited the description of viral-infected mice status in Results section and added the explain to address how the animals' suffering was minimized during this period in Methods section (Line 228-231, Page 5; Line 445-447, Page 12; Line 498-501, Page 14).
